DATA RELEASE

# Chromosomal-level genome assembly and single-nucleotide polymorphism sites of black-faced spoonbill *Platalea minor*

Hong Kong Biodiversity Genomics Consortium*,†

## ABSTRACT

*Platalea minor*, or black-faced spoonbill (Threskiornithidae), is a wading bird confined to coastal areas in East Asia. Due to habitat destruction, it was classified as globally endangered by the International Union for Conservation of Nature. However, the lack of genomic resources for this species hinders the understanding of its biology and diversity, and the development of conservation measures. Here, we report the first chromosomal-level genome assembly of *P. minor* using a combination of PacBio SMRT and Omni-C scaffolding technologies. The assembled genome (1.24 Gb) contains 95.33% of the sequences anchored to 31 pseudomolecules. The genome assembly has high sequence continuity with scaffold length N50 = 53 Mb. We predicted 18,780 protein-coding genes and measured high BUSCO score completeness (97.3%). Finally, we revealed 6,155,417 bi-allelic single nucleotide polymorphisms, accounting for ~5% of the genome. This resource offers new opportunities for studying the black-faced spoonbill and developing conservation measures for this species.

**Submitted:** 09 April 2024

\* Correspondence on behalf of the consortium: E-mail: jeromehui@cuhk.edu.hk

† Collaborative Authors: Entomological experts who validated the dataset and their affiliations appears at the end of the document

Preprint submitted at https://doi.org/10.1101/2024.04.08.588650

Included in the series: *Hong Kong Biodiversity Genomics* (https://doi.org/10.46471/GIGABYTE_SERIES_0006)

**Subjects** Genetics and Genomics, Animal Genetics, Ecology

## INTRODUCTION

The black-faced spoonbill *Platalea minor* (Threskiornithidae) (NCBI:txid259913, Figure 1A) is confined to coastal areas in East Asia, including Hong Kong, Macau, Taiwan, Vietnam, North Korea, South Korea, and Japan. The natural habitats of *P. minor* have been disturbed by human activities and industrialization, leading to the decline in the bird population over the last century [1, 2]. With an estimation of more than 6,000 individuals worldwide, the International Union for Conservation of Nature (also known as IUCN) has categorised the black-faced spoonbill as a globally endangered species. A quarter of the worldwide population of *P. minor* can be found in Hong Kong, and it is protected locally under the Wild Animals Protection Ordinance Cap 200. Genetic methods, including studies on genetic diversity and population structure, have been used to help retain this species with high conservation value [3, 4]. Nevertheless, a reference genome of this species was missing.

## METHODS
### Sample collection

Tissue samples of 14 *P. minor* individuals were collected from the north and northwestern parts of the New Territories, Hong Kong, between February 2015 and February 2020, with help from Kadoorie Farm and Botanic Garden. These samples were stored in 95% ethanol. Details of the sample collection are listed in Table 1.

**Figure 1.** (A) Picture of *Platalea minor*; (B) Statistics of the genome assembly generated in this study; (C) Hi-C contact map of the assembly visualised using Juicebox (v1.11.08); (D) Repetitive elements distribution.

## Isolation of high molecular weight genomic DNA

High molecular weight (HMW) genomic DNA was extracted from a single individual, labelled "BFS13". The tissue sample was first ground into a powder with liquid nitrogen and then processed with the Qiagen MagAttract HMW kit (Qiagen Cat. No. 67563), following the manufacturer's protocol. The final DNA sample was eluted with 120 µL of elution buffer (PacBio Ref. No. 101-633-500) and subjected to quality checks using the NanoDrop™ One/OneC Microvolume UV–Vis Spectrophotometer, Qubit® Fluorometer, and overnight pulse-field gel electrophoresis.

## DNA shearing, PacBio library preparation, and sequencing

Approximately 4.4 µg of HMW DNA was processed with DNA shearing through six centrifugation steps in a g-tube (Covaris Part No. 520079) at 2,000 × *g* for 2 min. The sheared DNA was transferred to a 2 mL DNA LoBind® Tube (Eppendorf Cat. No. 022431048) and stored at 4 °C. Overnight pulse-field gel electrophoresis was used to assess the fragment size distribution of the sheared DNA. Next, an SMRT bell library was constructed using the SMRTbell® prep kit 3.0 (PacBio Ref. No. 102-141-700), following the manufacturer's instructions. Briefly, the sheared DNA was processed with DNA repair, followed by polishing and tailing with A-overhang at both ends of each DNA strand. T-overhang SMRTbell adapters were then ligated to the polished ends to form SMRTbell templates, which were purified with SMRTbell® cleanup beads (PacBio Ref. No. 102158-300). The quantity and fragment size of the SMRTbell library were inspected with Qubit® Fluorometer and

**Table 1.** Summary of sequencing data.

| Sample | No. of reads | No. of bases | Coverage (×) | Accession number | KFBG reference number | Provenance details | Date specimen acquired at KFBG | Specimen type |
|---|---|---|---|---|---|---|---|---|
| *Sequencing data for reference genome* | | | | | | | | |
| PacBio HiFi | 2,707,085 | 25,352,990,722 | 20 | SAMN35152374 | K14352 | Mai Po | 15/02/2020 | Tissue in 95% ethanol |
| Omni-C | 518,587,164 | 77,788,074,600 | 63 | SAMN40731791 | B0181-2004 | Lok Ma Chau | 26/01/2004 | Tissue sample to be taken from carcass |
| *Population resequencing data* | | | | | | | | |
| BFS1 | 39,968,142 | 5,995,204,753 | 4.8 | SAMN35319659 | K7366 | Mai Po fishpond | 23/02/2015 | Tissue in 95% ethanol |
| BFS2 | 41,230,936 | 6,184,622,630 | 5.0 | SAMN35319660 | K7388 | Lok Ma Chau fishpond | 13/04/2015 | Portion of toe in 95% ethanol |
| BFS3 | 40,776,316 | 6,116,429,712 | 4.9 | SAMN35319661 | K8627 | Ma Cho Lung fishpond | 11/02/2016 | Tissue in 95% ethanol |
| BFS4 | 39,063,262 | 5,859,473,834 | 4.7 | SAMN35319662 | K11131 | Mai Po Gei Wai | 26/02/2018 | Tissue in 95% ethanol |
| BFS5 | 44,638,096 | 6,695,693,640 | 5.4 | SAMN35319663 | K11194 | Lut Chau, Nam San Wai | 10/04/2018 | Tissue in 95% ethanol |
| BFS6 | 41,890,130 | 6,283,502,065 | 5.1 | SAMN35319664 | K12542 | Lok Ma Chau pond | 18/12/2018 | Blood in 95% ethanol |
| BFS7 | 41,319,808 | 6,197,954,720 | 5.0 | SAMN35319665 | K12611 | Lok Ma Chau pond | 17/01/2019 | Tissue in 95% ethanol |
| BFS8 | 42,068,480 | 6,310,254,770 | 5.1 | SAMN35319666 | K12706 | Lok Ma Chau | 03/03/2019 | Blood in 95% ethanol |
| BFS9 | 42,539,460 | 6,380,900,668 | 5.1 | SAMN35319667 | K14133 | Lok Ma Chau | 26/11/2019 | Tissue in 95% ethanol |
| BFS10 | 40,359,264 | 6,053,871,746 | 4.9 | SAMN35319668 | K14208 | Lok Ma Chau | 25/12/2019 | Tissue in 95% ethanol |
| BFS11 | 41,496,386 | 6,224,440,405 | 5.0 | SAMN35319669 | K14366 | Mai Po | 20/02/2020 | Portion of feather in 95% ethanol |
| BFS12 | 39,361,374 | 5,904,189,803 | 4.8 | SAMN35319670 | K14401 | Tin Shui Wai Wetland Park | 12/03/2020 | Tissue in 95% ethanol |
| BFS14 | 41,319,546 | 6,197,915,065 | 5.0 | SAMN35319671 | K14324 | Tin Shui Wai Wetland Park | 03/02/2020 | Portion of feather in 95% ethanol |

overnight pulse-field gel electrophoresis, respectively. A nuclease treatment was conducted to remove any non-SMRTbell structures, and a subsequent size-selection step with 35% AMPure PB beads was used to remove short fragments. The final preparation of the library was performed using the Sequel® II binding kit 3.2 (PacBio Ref. No. 102-194-100). In brief, Sequel II primer 3.2 and Sequel II DNA polymerase 2.2 were added to anneal and bind to the SMRTbell templates, respectively. An internal control provided by the kit was also added. Finally, the library was loaded on the PacBio Sequel IIe System at an on-plate concentration of 90 pM with the diffusion loading mode. The sequencing was run in 30-h movies, with 120 min pre-extension. In total, one SMRT cell was used to output high-fidelity (HiFi) reads, and the sequencing data details are listed in Table 1.

## Omni-C library preparation and sequencing

An Omni-C library was constructed using the Dovetail® Omni-C® Library Preparation Kit (Dovetail Cat. No. 21005), following the manufacturer's protocol. A total of 80 mg of tissue was ground into a powder with liquid nitrogen, transferred to 1 mL 1× PBS, and then subjected to crosslinking with formaldehyde and digestion with endonuclease DNase I. An aliquot of 2.5 μL lysate was used for assessing lysate quantification and fragment size distribution using Qubit® Fluorometer and TapeStation D5000 HS Screen Tape, respectively. Then, end polishing, bridge ligation, and proximity ligation were carried out in the crosslinked DNA fragments. Next, crosslink reversal was performed, followed by DNA purification and size selection with SPRIselect™ Beads (Beckman Coulter Product No. B23317). The library preparation was continued with end repair and adapter ligation using the Dovetail™ Library Module for Illumina (Dovetail Cat. No. 21004), followed by DNA

purification with SPRIselect™ Beads. The DNA fragments were then captured with Streptavidin Beads and Universal and Index PCR Primers from the Dovetail™ Primer Set for Illumina (Dovetail Cat. No. 25005) were added to amplify the DNA library. A final size selection was carried out using SPRIselect™ Beads to retain DNA fragments ranging between 350 bp and 1000 bp. The quantity and fragment size distribution of the library were inspected by the Qubit® Fluorometer and the TapeStation D5000 HS ScreenTape, respectively. The final library was sequenced on an Illumina HiSeq-PE150 platform at Novogene. The details of the sequencing data are listed in Table 1.

## Genome assembly and gene model prediction

*De novo* genome assembly was performed using Hifiasm (RRID:SCR_021069) [5]. Haplotypic duplications were identified and removed using purge_dups (RRID:SCR_021173) based on the depth of HiFi reads [6]. Proximity ligation data from the Omni-C library was used to scaffold genome assembly by YaHS (RRID:SCR_022965) [7]. Transposable elements (TEs) were annotated using the automated Earl Grey TE annotation pipeline (version 1.2) as previously described [8]. Genome annotation was performed using Braker (v3.0.8) (RRID:SCR_018964) [9] with default parameters. Briefly, the genome was soft-masked using redmask (v0.0.2) [10]. A total of 2,468,534 aves reference protein sequences were downloaded from NCBI as protein references. A blood RNA-Seq dataset (SRR6650848) [11] was also downloaded from NCBI and aligned to the soft-masked genome using hisat2 (RRID:SCR_015530) [12] to generate the bam file. The protein and bam files were used as input to Braker for genome annotation.

## *Platalea minor* resequencing and single nucleotide polymorphism analysis

Genomic DNA from 13 *P. minor* individuals were isolated using the PureLink™ Genomic DNA Mini Kit (Invitrogen Cat no. K182002), following the manufacturer's instructions. The quality of DNA samples was assessed with the NanoDrop™ One/OneC Microvolume UV–Vis Spectrophotometer and 1% gel electrophoresis. Next, the samples were sent to Novogene for sequencing on an Illumina HiSeq-PE150 platform at approximately 6× coverage. The sequenced raw reads were then trimmed by Trimmomatic (v0.39, RRID:SCR_011848) [13] and cleaned with Kraken 2 (RRID:SCR_005484) [14]. The cleaned reads were aligned to large scaffolds (>500 kb, $n$ = 234), accounting for 97.1% of the *P. minor* reference genome, with BWA-MEM (RRID:SCR_022192) [15] using the parameters "-t 30 -M -R". Variant calling was performed using "HaplotypeCaller" and "GenotypeGVCFs" commands from the Genome Analysis Toolkit (GATK, RRID:SCR_001876, v4.1.2.0) [16]. Hard filtering was employed to filter out single nucleotide polymorphisms (SNPs) with the following criteria: quality by depth <2.0, Fisher strand bias >60.0, mapping quality <40.0, mapping quality rank sum test <−12.5, and read position rank sum test <−8.0. The remaining SNPs were further filtered for bi-allelic ("--min-alleles 2 --max-alleles 2"), no missing data ("--max-missing 1"), minimum summed site-depth (sumDP) of 20 and maximum sumDP of 130 to remove sites that were below one-third and above three-fold of the average sumDP, respectively, using the "--site-depth" and "--positions" options in VCFtools (v0.1.16, RRID:SCR_001235) [17]. The heterozygosity and inbreeding coefficient were estimated using VCFtools (v0.1.16) [17]. Details of the resequencing data are listed in Table 1.

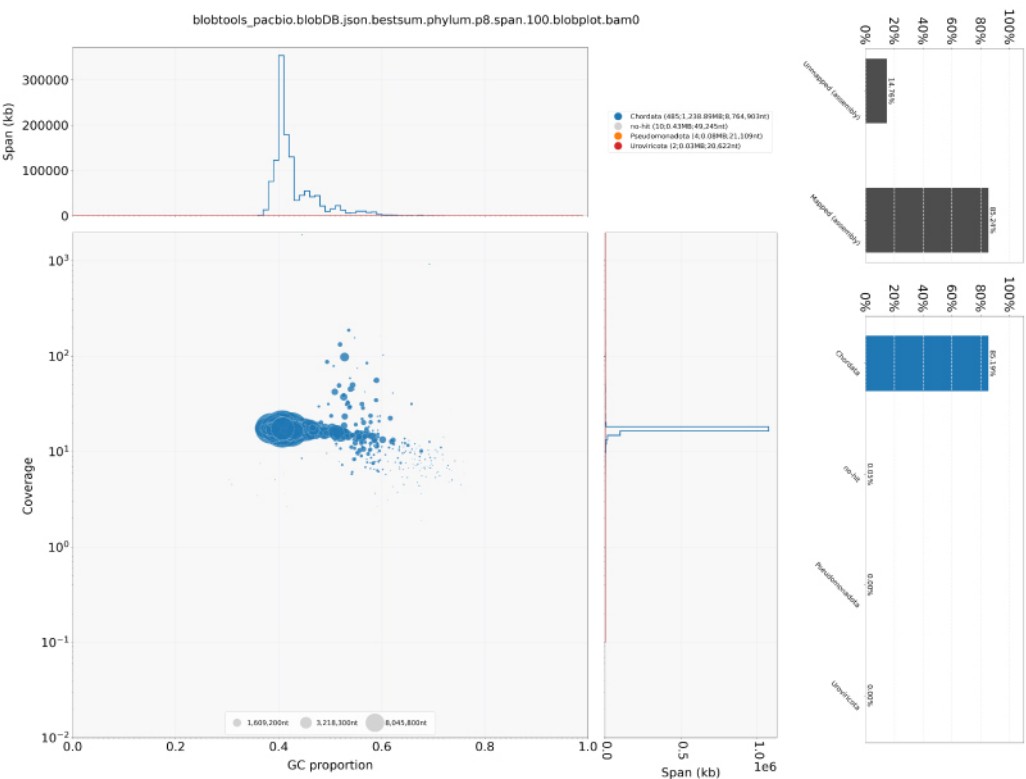

**Figure 2.** Genome assembly quality control and contaminant/cobiont detection. The upper panel shows the BlobPlot of the assembly. Each circle represents a scaffold with its size scaled according to its scaffold length, while the colour of the circle indicates the taxonomic assignment from BLAST similarity search results. The lower panel reveals the ReadCovPlot of the assembly, illustrating the proportion of unmapped and mapped sequences in the BLAST similarity search results on the left. The latter is further dissected according to the rank of phylum on the right.

## Data validation and quality control

During DNA extraction and PacBio library preparation, the samples were subjected to quality control with NanoDrop™ One/OneC Microvolume UV–Vis Spectrophotometer, Qubit® Fluorometer, and overnight pulse-field gel electrophoresis. The Omni-C library was inspected by Qubit® Fluorometer and TapeStation D5000 HS ScreenTape.

Regarding the genome assembly, the Hifiasm output was blast to the NT database, and the resulting output was used as input for Blobtools (v1.1.1, RRID:SCR_017618) [18]. Scaffolds identified as possible contaminations were removed from the assembly manually (Figure 2). A statistical kmer-based approach was applied to estimate the heterozygosity of the assembled genome. The repeat content and the corresponding sizes were analysed with k-mer 21 using Jellyfish (RRID:SCR_005491) [19] and GenomeScope (RRID:SCR_017014) [20] (Figure 3; Table 2). BUSCO (v5.5.0) [21] was used to assess the completeness of the genome assembly and gene annotation with a metazoan dataset (aves_odb10). HiC contact maps were generated using Juicer tools (version 1.22.01, RRID:SCR_017226) [22], following the Omni-C manual [23].

Omni-C reads and PacBio HiFi reads were used to measure the assembly completeness and the consensus quality (QV) using Merqury (v1.3, RRID:SCR_022964) [24] with kmer 20,

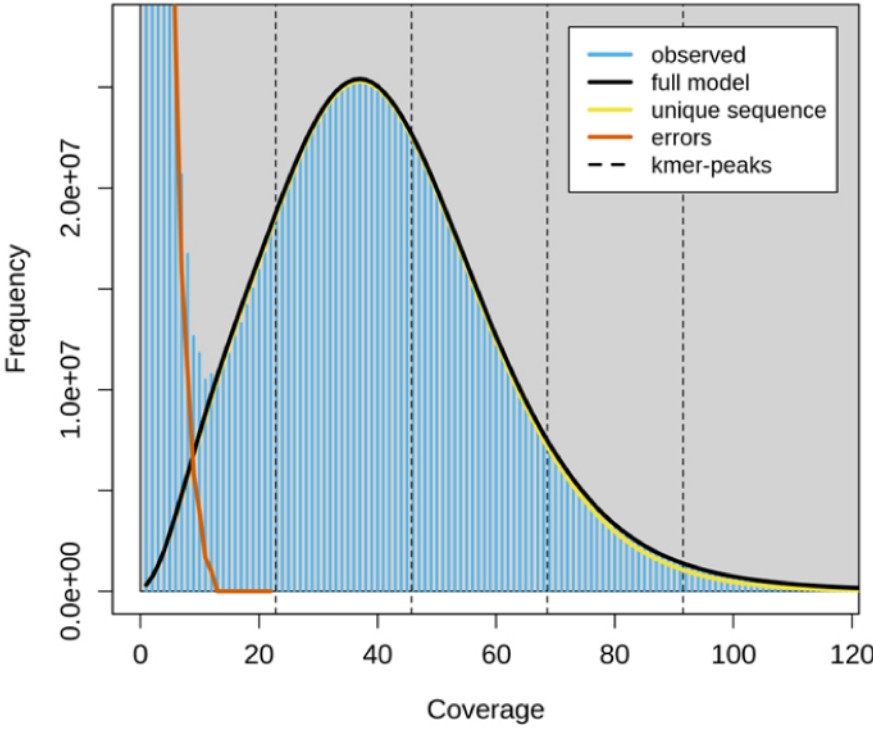

**Figure 3.** The GenomeScope profile with kmer 21.

**Table 2.** Summary of the GenomeScope statistics (*k* = 21).

| Property | min | max |
|---|---|---|
| Homozygous (aa) | 99.34% | 99.37% |
| Heterozygous (ab) | 0.63% | 0.66% |
| Genome Haploid Length (bp) | 1,141,324,739 | 1,144,280,536 |
| Genome Repeat Length (bp) | 112,880,770 | 113,173,108 |
| Genome Unique Length (bp) | 1,028,443,969 | 1,031,107,427 |
| Model Fit | 93.18% | 99.52% |
| Read Error Rate | 0.41% | 0.41% |

resulting in 95.0738% kmer completeness for the Omni-C data and 59.746 QV scores for the HiFi reads, corresponding to 99.999% accuracy.

The black-faced spoonbill genome assembly was also compared to five other avian genomes with chromosome-level assemblies and genome annotations, including *Gallus gallus* (Ggal: GCF_016699485.2), *Cuculus canorus* (Ccan: GCF_017976375.1), *Mycteria americana* (Mame: GCA_035582795.1), *Taeniopygia guttata* (Tgut: GCF_003957565.2), and *Theristicus caerulescens* (Tcae: GCA_020745775.1), which were downloaded from NCBI [25] and UCSC [26], respectively [27–29]. Macrosynteny was performed using MCScan (RRID:SCR_017650) with default parameters [30]. It is worth noting that some parts of the

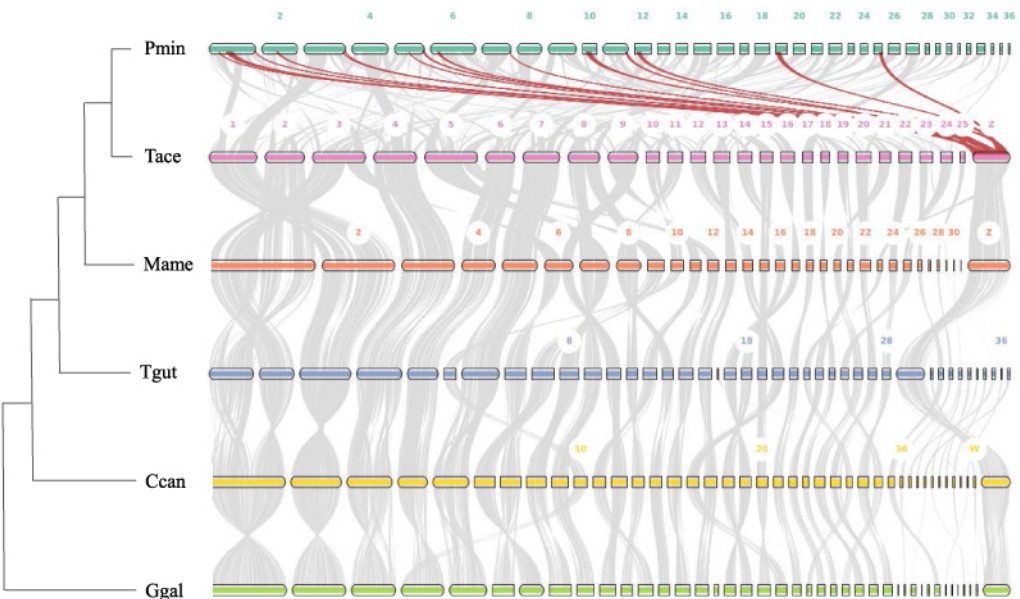

**Figure 4.** Macrosynteny between *P. minor* (Pmin), *Theristicus caerulescens* (Tcae), *Mycteria americana* (Mame), *Taeniopygia guttata* (Tgut), *Cuculus canorus* (Ccan), and *Gallus gallus* (Ggal).

largest scaffold in *P. minor* mapped to several *T. caerulescens* chromosomes, while genomes in other birds show relatively high syntenic conservation [31], which may warrant further investigation (Figure 4).

## RESULTS AND DISCUSSION

### Genome assembly of *P. minor*

A total of 25.35 Gb of HiFi bases was generated with an average HiFi read length of 9,365 bp with 20× data coverage (Table 1). After scaffolding with 77.79 Gb Omni-C sequencing data, the assembled genome size was 1.24 Gb in 468 scaffolds, with a scaffold N50 of 53 Mb and L50 of 8 (Tables 1, 3 and 4; Figures 1B and C). The genome size is comparable to those of other bird species in the family Threskiornithidae, which have genome sizes around 1.0–1.3 Gb, according to the data available in the NCBI Genbank, such as *Theristicus caerulescens* (1.20 Gb, GCA_020745775.1), *Nipponia nippon* (1.31 Gb, GCA_035839065.1), and *Mesembrinibis cayennensis* (1.19 Gb, GCA_013399675.1). The genome completeness was estimated by BUSCO (RRID:SCR_015008) with a value of 97.3% (aves_odb10) (Table 3; Figure 1B). The GC content was 42.98%. A total of 14,673 gene models were generated with 18,780 predicted protein-coding genes, having a mean coding-sequence length of 516 amino acids and a complete protein BUSCO value of 78.3% (Table 3).

### Repeat content

A total repeat content of 11.94% was found in the genome, which contained a lower level of repeat elements, similar to other avian genomes [32], with 2.49% unclassified elements. Of the remaining repeats, long interspersed nuclear elements (LINE) were the most abundant (5.10%), followed by long terminal repeats (LTR) (1.62%). In contrast, DNA, short interspersed nuclear elements (SINE), Penelope, and rolling circle were only present in low

**Table 3.** Genome statistics.

|  | **Platalea minor** |
|---|---|
| Total length (bp) | 1,239,504,613 |
| Number | 468 |
| Mean length (bp) | 2,648,514 |
| Longest scaffold length (bp) | 108,170,464 |
| Shortest scaffold length (bp) | 1,000 |
| N_count | 0.02% |
| Gaps | 998 |
| N50 | 53,081,851 |
| N50n | 8 |
| N70 | 31,851,659 |
| N70n | 15 |
| N90 | 14,707,644 |
| N90n | 25 |
| BUSCO (Geno, metazoa_odb10) | C:93.7%[S:93.4%,D:0.3%],F:1.6%,M:4.7%,n:954 |
| BUSCO (Geno, aves_odb10) | C:97.3%[S:97.0%,D:0.3%],F:0.5%,M:2.2%,n:8338 |
| Protein total length (amino acids) | 9,684,019 |
| Protein number (amino acids) | 18,780 |
| Protein mean length (amino acids) | 516 |
| BUSCO (Prot, metazoa_odb10) | C:88.4%[S:72.5%,D:15.9%],F:1.7%,M:9.9%,n:954 |
| BUSCO (Prot, aves_odb10) | C:78.3%[S:59.9%,D:18.4%],F:1.9%,M:19.8%,n:8338 |

proportions (DNA: 0.63%, SINE: 0.09%, Penelope: 0.06%, rolling circle: 0.02%). A complete catalogue of the repeat content of the genome can be found in Table 5 and Figure 1D.

## Single nucleotide polymorphism sites

A total of 6,046,878 bi-allelic SNPs were called from 13 *P. minor* individuals, accounting for ~0.5% of the genome. The mean individual heterozygosity was 0.142%. The lowest individual heterozygosity (0.077%) was close to other endangered bird species, such as *Pelecanus crispus* (0.60%) and *Nestor notabilis* (0.91%) [33]. The heterozygosity levels (0.108% to 0.116%) from five individuals were comparable to previous reports on spoonbills - black-faced spoonbill (0.101%–0.116%, mean 1.09%, *n* = 11) and royal spoonbill (0.098%–0.109%, mean 0.105%, *n* = 9) [4]. The remaining heterozygosity levels observed in this study were below the mean (0.221%) and median (0.213%) of heterozygosity reported from 40 avian species [33]. Signals of inbreeding were observed among the samples, with the inbreeding coefficient ($F_{IS}$) ranging from 0.331 to 0.720 (Table 6), providing additional evidence of a recent genetic bottleneck in the black-faced spoonbill population [4]. High levels of $F_{IS}$ have also been observed in other bird populations suffering from past bottlenecks [34]. These results highlighted the need for continuous efforts in monitoring *P. minor*.

## CONCLUSION AND REUSE POTENTIAL

This study presents the first chromosomal-level genome assembly and single-nucleotide polymorphism sites of black-faced spoonbill *Platalea minor*. These are useful and valuable resources for future population genomic studies aimed at better understanding spoonbill species numbers and conservation.

## DATA AVAILABILITY

The final assembly has been deposited at NCBI under the accession number JBBPFK000000000. The raw reads generated in this study, including Omni-C

**Table 4.** Scaffold information with a length larger than 1 Mb.

| Scaffold ID | Scaffold length (bp) | Cumulative % of the whole genome |
|---|---|---|
| scaffold_1 | 108,170,464 | 8.73% |
| scaffold_2 | 94,402,276 | 16.34% |
| scaffold_3 | 92,629,600 | 23.81% |
| scaffold_4 | 79,211,724 | 30.20% |
| scaffold_5 | 73,048,200 | 36.10% |
| scaffold_6 | 72,537,701 | 41.95% |
| scaffold_7 | 69,784,815 | 47.58% |
| scaffold_8 | 53,081,851 | 51.86% |
| scaffold_9 | 44,005,081 | 55.41% |
| scaffold_10 | 36,492,200 | 58.35% |
| scaffold_11 | 35,351,409 | 61.20% |
| scaffold_12 | 34,190,505 | 63.96% |
| scaffold_13 | 33,557,343 | 66.67% |
| scaffold_14 | 33,263,400 | 69.35% |
| scaffold_15 | 31,851,659 | 71.92% |
| scaffold_16 | 31,385,420 | 74.45% |
| scaffold_17 | 30,450,346 | 76.91% |
| scaffold_18 | 30,280,111 | 79.35% |
| scaffold_19 | 26,443,200 | 81.49% |
| scaffold_20 | 25,001,000 | 83.50% |
| scaffold_21 | 23,104,878 | 85.37% |
| scaffold_22 | 18,730,967 | 86.88% |
| scaffold_23 | 17,395,620 | 88.28% |
| scaffold_24 | 15,031,567 | 89.49% |
| scaffold_25 | 14,707,644 | 90.68% |
| scaffold_26 | 14,565,521 | 91.86% |
| scaffold_27 | 13,184,709 | 92.92% |
| scaffold_28 | 9,343,000 | 93.67% |
| scaffold_29 | 7,143,856 | 94.25% |
| scaffold_30 | 6,869,713 | 94.80% |
| scaffold_31 | 6,476,723 | 95.33% |
| scaffold_32 | 4,846,626 | 95.72% |
| scaffold_33 | 4,355,004 | 96.07% |
| scaffold_34 | 2,327,616 | 96.26% |
| scaffold_35 | 2,194,594 | 96.43% |
| scaffold_36 | 1,422,971 | 96.55% |
| scaffold_37 | 1,263,285 | 96.65% |
| scaffold_38 | 1,176,292 | 96.74% |
| scaffold_39 | 1,115,873 | 96.83% |

**Table 5.** Summary of the repetitive elements analysis.

| Classification | Coverage length (bp) | Count | Proportion (%) | No. of distinct classifications |
|---|---|---|---|---|
| DNA | 7,812,155 | 25,610 | 0.63 | 3,780 |
| LINE | 63,189,657 | 102,036 | 5.10 | 5,652 |
| LTR | 20,129,006 | 29,922 | 1.62 | 3,928 |
| Other (Simple Repeat, Microsatellite, RNA) | 23,864,680 | 3,875 | 1.93 | 1,118 |
| Penelope | 690,955 | 1,966 | 0.06 | 646 |
| Rolling Circle | 287,526 | 744 | 0.02 | 404 |
| SINE | 1,165,920 | 4,910 | 0.09 | 885 |
| Unclassified | 30,908,397 | 60,141 | 2.49 | 5,695 |
| **SUM** | **148,048,296** | **229,204** | **11.94** | **22,108** |



**Table 6.** Number of SNPs, statistics of heterozygosity and inbreeding coefficient of 13 *Platalea minor* individuals.

| Sample ID | No. of sites with observed heterozygosity | $H_0$ (%) | $F_{IS}$ |
|---|---|---|---|
| BFS1 | 1,015,586 | 0.168 | 0.382 |
| BFS2 | 673,637 | 0.111 | 0.590 |
| BFS3 | 654,457 | 0.108 | 0.602 |
| BFS4 | 1,013,752 | 0.168 | 0.383 |
| BFS5 | 700,877 | 0.116 | 0.573 |
| BFS6 | 1,076,231 | 0.178 | 0.345 |
| BFS7 | 1,080,566 | 0.179 | 0.342 |
| BFS8 | 1,055,650 | 0.175 | 0.357 |
| BFS9 | 463,725 | 0.077 | 0.718 |
| BFS10 | 997,870 | 0.165 | 0.393 |
| BFS11 | 700,069 | 0.116 | 0.574 |
| BFS12 | 652,122 | 0.108 | 0.603 |
| BFS14 | 1,109,354 | 0.183 | 0.325 |

(SAMN40731791) and PacBio HiFi (SAMN35152374) data, have been deposited in the NCBI database under the BioProject accession number PRJNA973839. The genome, genomic and repeat annotation files have been deposited and are publicly available in Figshare [35].

## ABBREVIATIONS

HiFi, high-fidelity; HMW, high molecular weight; LINE, long interspersed nuclear element; LTR, long terminal repeat; QV, consensus quality; SINE, short interspersed nuclear element; SNP, single nucleotide polymorphisms; sumDP, summed site-depth; TE, transposable elements.

## DECLARATIONS

### Ethics approval and consent to participate

The authors declare that ethical approval was not required for this type of research.

### Competing interests

The authors declare that they do not have competing interests.

### Authors' contributions

JHLH, TFC, LLC, SGC, CCC, JKHF, JDG, SCKL, YHS, CKCW, KYLY and YW conceived and supervised the study; WLS carried out DNA extraction, library preparation and sequencing; WN performed genome assembly and gene model prediction; STSL carried out the SNPs calling and Fst calculations; PC, AL, LRJ and HYY collected and maintained the samples. All authors approved the final version of the manuscript.

### Funding

This work was funded and supported by the Hong Kong Research Grant Council Collaborative Research Fund (C4015-20EF), CUHK Strategic Seed Funding for Collaborative Research Scheme (3133356) and CUHK Group Research Scheme (3110154).

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

## DETAILS OF COLLABORATIVE AUTHORS

### • List of authors in Hong Kong Biodiversity Genomics Consortium

Jerome H. L. Hui,[1] Ting Fung Chan,[2] Leo Lai Chan,[3] Siu Gin Cheung,[4] Chi Chiu Cheang,[5,6] James Kar-Hei Fang,[7] Juan Diego Gaitan-Espitia,[8] Stanley Chun Kwan Lau,[9] Yik Hei Sung,[10,11] Chris Kong Chu Wong,[12] Kevin Yuk-Lap Yip,[13,14] Yingying Wei,[15] Wai Lok So,[1] Wenyan Nong,[1] Sean Tsz Sum Law,[1] Paul Crow,[16] Aiko Leong,[16] Liz Rose-Jeffreys,[16] Ho Yin Yip[1]

[1]Institute of Environment, Energy and Sustainability, The Chinese University of Hong Kong, Simon F.S. Li Marine Science Laboratory, State Key Laboratory of Agrobiotechnology, Hong Kong SAR, China

[2]State Key Laboratory of Agrobiotechnology, The Chinese University of Hong Kong, Hong Kong SAR, China

[3]State Key Laboratory of Marine Pollution and Department of Biomedical Sciences, City University of Hong Kong, Hong Kong SAR, China

[4]State Key Laboratory of Marine Pollution and Department of Chemistry, City University of Hong Kong, Hong Kong SAR, China

[5]Department of Science and Environmental Studies, The Education University of Hong Kong, Hong Kong SAR, China

[6]EcoEdu PEI, Charlottetown, PE, C1A 4B7, Canada

[7]Research Institute for Future Food, and State Key Laboratory of Marine Pollution, The Hong Kong Polytechnic University, Hong Kong SAR, China

[8]The Swire Institute of Marine Science and School of Biological Sciences, The University of Hong Kong, Hong Kong SAR, China

[9]Department of Ocean Science, The Hong Kong University of Science and Technology, Hong Kong SAR, China

[10]Science Unit, Lingnan University, Hong Kong SAR, China

[11]School of Allied Health Sciences, University of Suffolk, Ipswich, IP4 1QJ, UK

[12]Croucher Institute for Environmental Sciences, and Department of Biology, Hong Kong Baptist University, Hong Kong SAR, China

[13]Department of Computer Science and Engineering, The Chinese University of Hong Kong, Hong Kong SAR, China

[14]Sanford Burnham Prebys Medical Discovery Institute, CA, La Jolla, USA

[15]Department of Statistics, The Chinese University of Hong Kong, Hong Kong SAR, China

[16]Kadoorie Farm and Botanic Garden, Lam Kam Road, Tai Po, New Territories, Hong Kong SAR, China

