## [Editor Report]

Editor’s AssessmentThis work is part of a series of papers from the Hong Kong Biodiversity Genomics Consortium sequencing the rich biodiversity of species in Hong Kong (see https://doi.org/10.46471/GIGABYTE_SERIES_0006). This example assembles the genome of the black-faced spoonbill (Platalea minor), an emblematic wading bird from East Asia that is classified as globally endangered by the IUCN. This Data Release reporting a 1.24Gb chromosomal-level genome assembly produced using a combination of PacBio SMRT and Omni-C scaffolding technologies. BUSCO and Merqury validation were carried out, gene models created, and peer reviewers also requested MCscan synteny analysis. This showed the genome assembly had high sequence continuity with scaffold length N50=53 Mb. Presenting data from 14 individuals this will hopefully be a useful and valuable resources for future population genomic studies aimed at better understanding spoonbill species numbers and conservation.Editor’s AssessmentThis work is part of a series of papers from the Hong Kong Biodiversity Genomics Consortium sequencing the rich biodiversity of species in Hong Kong (see https://doi.org/10.46471/GIGABYTE_SERIES_0006). This example assembles the genome of the black-faced spoonbill (Platalea minor), an emblematic wading bird from East Asia that is classified as globally endangered by the IUCN. This Data Release reporting a 1.24Gb chromosomal-level genome assembly produced using a combination of PacBio SMRT and Omni-C scaffolding technologies. BUSCO and Merqury validation were carried out, gene models created, and peer reviewers also requested MCscan synteny analysis. This showed the genome assembly had high sequence continuity with scaffold length N50=53 Mb. Presenting data from 14 individuals this will hopefully be a useful and valuable resources for future population genomic studies aimed at better understanding spoonbill species numbers and conservation.

---

## [Reviewer Report]

Indicate in the comments box below whether you are happy with the changes made or if the manuscript is unacceptable.Comments on revised manuscriptThe authors incorporated the revisions nicely and have produced a quality manuscript. Well done. Minor revisions Line 46: A comma is needed after (Threskiornithidae). Line 47: “The” should not be capitalized. Line 48: This should read “as a globally endangered species.” Line 49: “However, the lack of genomic resources for the species hinders the understanding of its biology…” Line 56: Consider changing “also revealed” to “identified” to avoid repetition from the previous sentence. Line 65: Insert “the” before “bird’s.” Lines 69-70: Move “locally” higher in the sentence – “and it is protected locally…” Line 72: Replace “as of to date” with “prior to this study”. Lines 78-79: Pluralize “part.” Line 86: Replace “proceeded” with “processed.” Line 133: “…are listed in Table 1.” Line 158: “accounted” Line 159: “Variant calling was performed using…” Line 161: “Hard filtering was employed…” Lines 200-201: “The heterozygosity levels… from five individuals were comparable to previous reports on spoonbills – black-faced spoonbill … and royal spoonbill … (Li et al. 2022).” Line 202: New sentence. “The remaining heterozygosity levels observed…” Line 206: “…genetic bottleneck in the black-faced spoonbill…” Lines 208-209: “These results highlight the need…” Lines 213-214: “…which are useful and precious resources for future population genomic studies aimed at better understanding spoonbill species numbers and conservation.” Line 226: Missing a period after “heterozygosity.” For references, consider adding DOIs. Some citations have them but most citations would benefit from this addition.

---

## [Reviewer Report]

Indicate in the comments box below whether you are happy with the changes made or if the manuscript is unacceptable.Comments on revised manuscriptI previously reviewed this manuscript and overall the authors have done a nice job addressing all of my comments. I appreciate that the authors include the MCscan analysis that I suggested. However, the alignment of the P. minor assembly and annotations to other genomes suggests rampant mis-assembly or translocations. Birds have fairly high synteny and I would expect Pmin to look more similar to the comparison between T. caerulescens and M. americana in the MCscan plot. For instance, parts of the largest scaffold in the Pmin assembly map to multiple different chromosomes in the Tcae assembly. Similarly, the Z in Tcae maps to 11 different scaffolds in the Pmin assembly and there does not appear to be a single large scaffold in the Pmin assembly that corresponds to the Z chromosome. The genome seems to be otherwise of strong quality, so I urge the authors to double-check their MCscan synteny analysis. If this pattern remains, can you please add some comments about it to the end of the Data Validation and Quality Control section? I think other readers will also be surprised at the low levels of synteny apparent between the spoonbill and ibis assemblies.

---

## [Reviewer Report]

Reviewer name and names of any other individual's who aided in reviewer Richard Flamio Jr. Do you understand and agree to our policy of having open and named reviews, and having your review included with the published papers. (If no, please inform the editor that you cannot review this manuscript.)YesIs the language of sufficient quality?NoPlease add additional comments on language quality to clarify if needed
There are some grammatical errors and spelling mistakes throughout the text. Are all data available and do they match the descriptions in the paper? YesAdditional CommentsAre the data and metadata consistent with relevant minimum information or reporting standards? See GigaDB checklists for examples <a href="http://gigadb.org/site/guide" target="_blank">http://gigadb.org/site/guide</a>YesAdditional CommentsIs the data acquisition clear, complete and methodologically sound?YesAdditional CommentsIs there sufficient detail in the methods and data-processing steps to allow reproduction?YesAdditional CommentsThe authors did a phenomenal job at detailing the methods and data-processing steps. Is there sufficient data validation and statistical analyses of data quality? YesAdditional CommentsIs the validation suitable for this type of data?YesAdditional CommentsIs there sufficient information for others to reuse this dataset or integrate it with other data?YesAdditional CommentsAny Additional Overall Comments to the AuthorVery nice job on the paper. The methods are sound and the statistics regarding the genome assembly are thorough. My only two comments are: 1) I think the paper could be improved by the correction of grammatical errors, and 2) I am interested in a discussion about the number of chromosomes expected for this species (or an estimate) based on related species and if the authors believe all of the chromosomes were identified. For example, is the karyotype known or can the researchers making any inferences about the number of microchromosomes in the assembly? Please see a recent paper I wrote on microchromosomes in the wood stork assembly (https://doi.org/10.1093/jhered/esad077) for some ideas in defining the chromosome architecture of the spoonbill and/or comparing this architecture to related species. RecommendationMajor Revision

---

## [Reviewer Report]

Reviewer name and names of any other individual's who aided in reviewer Phred BenhamDo you understand and agree to our policy of having open and named reviews, and having your review included with the published papers. (If no, please inform the editor that you cannot review this manuscript.)YesIs the language of sufficient quality?YesPlease add additional comments on language quality to clarify if needed
Generally yes, the language is sufficiently clear. However, a number of places could be refined and extra words removed.Are all data available and do they match the descriptions in the paper? NoAdditional CommentsAdditional data is available on fig share.   I do not see any of the tables that are cited in the manuscript and contain legends. Am I missing something. Also there is no legend for the GenomeScope profile in figure 3.  The assembly appears to be on genbank as a scaffold level assembly, can you list this accession info in the data availability section in addtion to the project number..Are the data and metadata consistent with relevant minimum information or reporting standards? See GigaDB checklists for examples <a href="http://gigadb.org/site/guide" target="_blank">http://gigadb.org/site/guide</a>YesAdditional CommentsIs the data acquisition clear, complete and methodologically sound?YesAdditional CommentsIs there sufficient detail in the methods and data-processing steps to allow reproduction?YesAdditional CommentsIs there sufficient data validation and statistical analyses of data quality? NoAdditional CommentsOverall fine, but some additional analyses would aid the paper. Comparison of the spoonbill genome to other close relatives using a synteny plot would be helpful.  It would also be useful to put heterozygosity and inbreeding coefficients into context by comparing to results from other species. Is the validation suitable for this type of data?YesAdditional CommentsIs there sufficient information for others to reuse this dataset or integrate it with other data?YesAdditional CommentsAny Additional Overall Comments to the AuthorHui et al. report a chromosome level genome for the black-faced spoonbill, a endangered species of coastal wetlands in East Asia. This genome will serve as an important genome for understanding the biology of and conserving this species.   Generally, the methods are sound and appropriate for the generation of genomic sequence.   Major comments: This is a highly contiguous genome in line with metrics for Vertebrate Genomics Project genomes and other consortia. The authors argue that they have assembled 31 Pseudo-molecules or chromosomes. It would be nice to see a plot showing synteny of these 31 chromosomes and a closely related species with a chromosome level assembly (e.g. Theristicus caerulescens; GCA_020745775.1)  The tables appear to be missing from the submitted manuscript?  Minor comments: Line 49: delete its  Line 49-51: This sentence is a little awkward, please revise.  Line 64: delete 'the'  Line 67: replace 'with' with 'the spoonbil as a'  Line 68: delete 'Interestingly'  Line 70: can you be more specific about what kind of genetic methods had previously been performed?  Line 79: can you provide any additional details on the necessary permits and/or institutional approval  Line 78: what kind of tissue? or were these blood samples?  Line 110: do you mean movies?  Line 143: replace data with dataset  Line 163: it may be worth applying some additional filters in vcftools, e.g. minor allele freq., min depth, max depth, what level of missing data was allowed?, etc.  Line 171: delete 'resulted in'  Line 172: do you mean scaffold L50 was 8? Line 191-195: some context would be useful here, how does this level of heterozygosity and inbreeding compare to other waterbirds?  Line 217: why did you use the Metazoan database and not the Aves_odb10 database for Busco?  Figure 1b: Number refers to what, scaffolds? Be consistent with capitalization for Mb. It seems like the order of scaffold N50 and L50 were reversed.   Figure 3 is missing a legend.
RecommendationMajor Revision